# Clinical and Immunologic Characteristics of Non-Hematologic Cancers in Patients with Inborn Errors of Immunity

**DOI:** 10.3390/cancers15030764

**Published:** 2023-01-26

**Authors:** Samaneh Delavari, Yating Wang, Tannaz Moeini shad, Salar Pashangzadeh, Farzad Nazari, Fereshte Salami, Hassan Abolhassani

**Affiliations:** 1Research Center for Immunodeficiencies, Pediatrics Center of Excellence, Children’s Medical Center, Tehran University of Medical Science, 1419733151 Tehran, Iran; 2Division of Clinical Immunology, Department of Biosciences and Nutrition, Karolinska Institute, 14183 Stockholm, Sweden

**Keywords:** inborn errors of immunity, primary immunodeficiency, malignancy, solid tumors

## Abstract

**Simple Summary:**

Comprehensive studies on non-hematologic tumors in patients with inborn errors of immunity are scarce. Using a well-established registry of patients with long-term follow-up, molecular defects associated with these cancers were reported for the first time among these patients. Using the retrospective data available of this national registry of patients with primary immune defects, we clarified that almost all cancer hallmarks are involved in the development of non-hematologic cancers in patients presenting with non-hematologic cancers.

**Abstract:**

Inborn errors of immunity (IEI) are a heterogeneous group of inherited disorders, and almost 500 genes associated with these disorders have been identified. Defects in IEI genes lead to diverse clinical manifestations including increased susceptibility to recurrent or prolonged infections, immune dysregulation phenotypes (such as severe atopy, allergy, autoimmunity, and uncontrolled inflammation, lymphoproliferation), as well as predisposition to malignancies. Although the majority of IEI patients present hematologic cancers, the characteristics of other types of cancers are not well described in these groups of patients. By investigating 5384 IEI patients registered in the Iranian national registry the clinical and immunologic phenotypes of patients with non-hematologic cancers were compared with other malignant and non-malignant patients. Solid tumors were reported <20% of malignant IEI patients (*n* = 27/144 patients) and appeared to be very heterogeneous by type and localization as well as molecular defects (mainly due to DNA repair defect resulted from ATM deficiency). The correlation between the type of malignancy and survival status was remarkable as patients with non-hematologic cancers survive higher than IEI patients with hematologic cancers. Our findings showed that different types of malignancy could be associated with specific entities of IEI. Therefore, the education of physicians about the risk of malignancies in IEI is required for personalized treatment and appropriate management of patients.

## 1. Introduction

Inborn errors of immunity (IEI), previously referred to as primary immunodeficiency disorders constitute heterogeneous genetic diseases, commonly presented with infections or other immune-related manifestations including malignancies [1,2]. Thanks to the improvements in molecular diagnosis methodologies, approximately 500 genes associated with IEI have been described affecting both innate and adaptive immunity [3]. These genetic alterations result in developmental arrest and/or the functional defects of different immune system components [4]. Among all reported IEI patients in the world, <15% of patients received a final molecular diagnosis demonstrating the essential need for clarification of pathogenic mechanisms in the remaining cases [5].

IEI can lead to increased morbidity and mortality in affected cases [6]. Malignancies have been reported as one of the leading mortality causes associated with IEI, especially due to hematological cancers when infectious complications of these patients can be managed adequately [2,7]. Since IEIs are very uncommon, it is very difficult to establish the exact molecular background of malignancies correlated with cancers. Nonetheless, the development of cancers has been reported in less than one-fourth of IEI patients and it is estimated that a minority of patients can present solid non-hematologic tumors [8]. The most important reported mechanism of malignancy is avoiding immune destruction (defective tumor immune surveillance to detect and eliminate nascent transformed cells) [9,10]. However, almost all other cancer hallmarks are commonly associated with IEI including genome instability, and mutation, enabling replicative immortality, tumor-promoting inflammation, resisting cell death, sustaining proliferative signaling, evading growth suppressors, deregulating cellular energetics, inducing angiogenesis, activating invasion and metastasis, unlocking phenotypic plasticity, senescent cells, non-mutational epigenetic reprogramming, and polymorphic microbiomes [9,11].

The type of malignancy in IEI patients is highly dependent on the genetic defects and their functional activity [2,12,13,14,15]. Based on a comprehensive analysis of large patient cohorts, lymphoma (non-Hodgkin’s and Hodgkin’s) and leukemia account for 70% and 10% of the malignancies in patients with IEI [12,16]. However other forms of solid tumors also can be identified in higher prevalence in these patients compared to normal population, especially gastric adenocarcinoma [16,17,18]. In the current study, we aimed to compare the clinical and immunologic characteristics of IEI patients with non-hematologic and hematologic cancers and investigate their underlying molecular defects.

## 2. Materials and Methods

### 2.1. Patients

A total of 5384 patients with IEI were registered in the Iranian national IEI database between the years 1999 and 2020 [19,20]. To have a more comprehensive investigation of patients with malignancy, this retrospective longitudinal study was conducted between August 2022 and October 2022 among registered patients to analyze the clinical, immunologic, and molecular data. After establishing a research design, the study was approved by the ethics committee of Tehran University of Medical Science, Tehran, Iran. Patients with incomplete data or those who did not meet the criteria were excluded. The diagnosis of IEI was carried out based on the Middle East and North Africa (MENA) and European Society for Immunodeficiency (ESID) diagnostic criteria [21,22]. The patients were categorized into IEI subcategories using the International Union of Immunological Societies (IUIS) classification [3]. Written informed consent was obtained from all subjects involved in the study or their parents.

### 2.2. Clinical Evaluation in IEI Patients

We gathered all IEI patients’ medical relevant data using a comprehensive questionnaire that included demographic data, clinical complications, immunological and paraclinical tests covering the diagnostic basic lab tests at the time of the first presentation until specific diagnostic evaluation for confirmation of cancer, and during and after cancer treatment. For each patient, malignant presentations before and/or after IEI diagnosis were documented. The accurate diagnosis of cancer was based on international criteria considering medical, imaging, biochemical and histopathological documents as described previously [10,14,15]. The evaluation for cancer diagnosis was re-assessed for all IEI cases by an oncologist and a subspecialist related to the involved tissue/organ and was matched with topography and morphology codes obtained from the International Classification of Diseases, 10th revision (ICD-10). Due to the aim of the study, if these patients had concomitant non-hematologic and hematologic cancers, it was classified in the group of non-hematologic cancers but also reported in detail separately.

### 2.3. Genetic Investigation and Diagnoses in IEI Patients

Targeted sequencing was conducted on extracted genomic DNA from a selected group of IEI cases with a classical clinical presentation suggestive of a specific entity and agreed to genetic testing as described previously [3,23,24,25,26,27]. Whole-exome sequencing was performed to detect single nucleotide variants insertion/deletions and large deletions for patients in whom targeted sequencing failed or had medical features resembling several genetic defects. Candidate genetic alterations were weighed based on the American College of Medical Genetics and Genomics criteria including considering the allele frequency in the population, computational data, immunological data and clinical phenotyping. Only variants with pathologic and likely pathogenic scores and correct Mendelian inheritance pattern were included as the final molecular diagnosis [28,29,30,31].

After confirmation of their clinical and genetic diagnosis patients were classified according to the International Union of Immunological Societies (IUIS) updated classification. This classification includes nine categories of immunodeficiencies affecting cellular and humoral immunity (non-syndromic combined immunodeficiency or CID), combined immunodeficiencies with associated or syndromic features (syndromic CID and bone marrow failure), predominantly antibody deficiencies (PAD), diseases of immune dysregulation, congenital defects of phagocyte number or function (phagocytic disorders), autoinflammatory disorders, defects in intrinsic and innate immunity, complement deficiencies, bone marrow failure and phenocopies of inborn errors of immunity [3].

### 2.4. Statistical Analysis

Data were analyzed using the SPSS statistical software package version 25.0 (IBM corporation, Chicago, IL, USA) and R statistical systems (version 3.4.1., R Foundation for Statistical Computing, Vienna, Austria). The Shapiro-Wilk test was used to validate the assumption of normality for a variable, and the nonparametric or parametric tests were carried out according to the normality assumption. A p-value < 0.05 was considered statistically significant.

## 3. Results

Altogether 144 patients (2.6% of the total 5384 patients in IEI registry) with a definite diagnosis of malignancy were recruited for this study, from which 41.6% were female (Table 1). At the time of this study, the median age of cancer patients was 23.2 (interquartile range, IQR 11.0–28.5) years which was significantly higher than the median age of the IEI patients without cancer (9.5 (1.5–37.0) years, *p* < 0.001). Among these IEI patients with cancer presentation, hematologic malignancies were the most common type of malignancy expectedly. However, non-hematologic malignancies constituted 18.7% of patients (*n* = 27). Of note, the main non-hematologic malignancies included gastrointestinal cancers (*n* = 10, 8 gastric adenocarcinomas, 2 colorectal cancer), head/neck cancers (*n* = 8, 4 squamous cell carcinoma of tongue, 2 mandibular squamous cell carcinoma, 2 brain tumor), skin cancers (*n* = 3, 2 squamous cell carcinoma and 1 melanoma), breast cancers (*n* = 3, 2 invasive ductal carcinomas, 1 in situ ductal carcinoma), ovarian cancers (*n* = 2, both ovarian cystadenoma), and thyroid cancer (*n* = 1, papillary thyroid cancer). The correlation between the type of malignancy and IEI clinical features was remarkable as patients with non-hematologic cancers present a higher rate of chronic lung complications including bronchiectasis than IEI patients with hematologic cancers (40.7% vs. 19.6%, *p* = 0.02). Of note, chronic enteropathy was also among major-specific gastrointestinal manifestations observed significantly more frequently in patients with non-hematologic cancers (40.7% vs. 11.9%, *p* < 0.001). This association with chronic enteropathies were even higher in patients with gastrointestinal cancers compared to other non-hematologic cancers (9/10 vs. 1/17, *p* = 005). In general, patients with non-hematologic cancers were predisposed less than hematologic cancers to severe infection, autoimmunity and lymphoproliferative disorders, but these differences were not statistically significant (Table 1). Expectedly, the age of diagnosis of malignancy were correlated with the current age of patients at the time of study (r = 0.65. *p* = 0.03).

Patients with a clinical diagnosis of CID and immune dysregulation presented exclusively with hematologic malignancies. However, non-hematologic malignancies were recorded in patients with PAD (*n* = 14), syndromic CID (*n* = 8) and defects in intrinsic and innate immunity (*n* = 5). No tumor was documented in IEI cases with phagocyte disorders and autoinflammatory disease and complement deficiency. Among PAD patients, the most prevalent type associated with non-hematologic malignancy was clinically diagnosed as common variable immunodeficiency (10 patients), selective IgA deficiency (3 patients) and hyper IgM syndrome (1 patient). Regarding patients with the syndromic CID group, Ataxia-Telangiectasia and hyper IgE syndrome patients had solid tumors in 7 cases and 1 cases, respectively. In patients with inborn errors of innate immunity cases of Chronic mucocutaneous candidiasis (*n* = 3) and Mendelian susceptibility to mycobacterial diseases (*n* = 2) were reported with non-hematologic malignancies. General immunologic profiles of patients in both groups are presented in Table 2.

Genetic evaluations were conducted in all malignant patients in this cohort (selected 15 patients with agammaglobulinemia and ataxia-telangiectasia using targeted sequencing and remaining 129 patients using WES) and molecular diagnosis were confirmed in 78 patients (54.1%) from which 18 different IEI genes were identified as the underlying pathogenesis. Considering the genetic diagnostic yield among 27 IEI patients with non-hematologic malignancies, the molecular diagnosis was identified in 16 (59.2%) which in comparison with IEI cases associated with hematologic malignancies were slightly higher but not significantly different (62/117, 52.9%, *p* = 0.31). These deleterious variants were affecting mainly the DNA repair pathway (ATM deficiency all presented with clinical presentation of ataxia-telangiectasia (*n* = 21)), T cell development (AIRE deficiency (*n* = 5), RAG1 deficiency (*n* = 2), DOCK8 deficiency (*n* = 5), STK4 deficiency (*n* = 4), STAT3 deficiency (*n* = 3), STAT1 gain-of-function (*n* = 4)), B cell development (BTK deficiency (*n* = 12), IGLL1 deficiency (*n* = 1), IKZF1 deficiency (*n* = 2), TNFRSF13B deficiency (*n* = 2), PI3KR1 deficiency (*n* = 4), PI3KCD gain-of-function (*n* = 2)), T/B cell co-stimulation with increased susceptibility to EBV (CD27 deficiency (*n* = 3), CD70 deficiency (*n* = 3), RASGRP1 deficiency (*n* = 1)), NK cell development (CD16 deficiency (*n* = 2)) innate immune signaling (IL12RB1 deficiency (*n* = 2)) (Figure 1).

Our findings highlighted the specific IEI genes underlying non-hematologic malignancies including ATM deficiency (*n* = 7), AIRE deficiency (*n* = 2), IKZF1 deficiency (*n* = 1), PI3KR1 deficiency (*n* = 2), IL12RB1 deficiency (*n* = 2), and STAT1 gain-of-function (*n* = 2). Of note, overlap phenotypes of both hematologic and non-hematologic malignancies were documented in 3 patients with *ATM* mutations. Moreover, in our IEI patient cohort defects in the *IL12RB1* gene were exclusively presented with non-hematologic cancers and no individual with hematologic cancers was detected (Table 3).

From 144 patients, 48 IEI cases with malignancy (33.3%) were deceased (22 females). Among these deceased patients detailed cause of death were recorded in 39 individuals indicating cancer-related mortality in 34 cases (mainly due to hemorrhagic and thromboembolic phenomena, organ invasion and end-organ failure, respiratory or cardiovascular insufficiency, cancer-treatment side-effects, and cachexia) and non-cancer related causes in 5 patients (mainly due to prominent infectious complications and sepsis). We compared the survival of non-hematologic malignant patients was significantly better than individuals with hematologic cancers (4/27 vs. 44/117, *p* = 0.02). Moreover, patients in the non-hematologic malignant group significantly survived longer as their current age at the time of the study was 27.4 (IQR 22.7–36.4) years compared to 17.0 (IQR 10.9–24.2) years in a hematologic malignant group (*p* = 0.01). Similar observations were confirmed with survival analysis using Kaplan-Meier analysis (log-rank test *p* = 0.02, Figure 2). To compare the association between survival status and molecular pathogenesis, we investigated genetic defects with both non-hematologic and hematologic malignancies. Surprisingly all deceased patients from non-hematologic malignancy were ATM-deficient patients (*n* = 4, 100%) including all 3 cases with overlapping hematologic cancers. The age at the time of death in these 3 patients was not significantly different from other ATM-deficient patients with only- hematologic cancers (13.5 (IQR 8.6–19.5) vs. 14.7 (IQR 10.0–21.5) years, *p* = 0.62).

## 4. Discussion

In the current retrospective study, we evaluated for the first time the overall incidence of non-hematologic malignancy in IEI patients enrolled in the national registry in detail. We observed almost 20% of malignant patients with documented non-hematologic cancers, consisting of a prevalence of 500 per 100,000 (mainly due to gastrointestinal cancers, head/neck cancers and breast cancers) at the median age of 27.4 years. The prevalence of cancers in the normal population (age-standardized rate 300 per 100,000, mainly due to breast cancers, gastrointestinal cancer and lung cancers) indicates almost twice the higher risk of developing non-hematologic malignancies in our cohort [32,33]. Of note, this risk still is far lower than the predisposition of IEI patients to hematologic malignancy, presenting almost 20-fold higher risk mainly to non-Hodgkin’s lymphoma. In contrast in the normal population hematologic malignancies are mainly due to leukemia with a prevalence of 10 per 100,000. The age-standardized mortality rate of cancer in the normal Iranian population was also 80 per 100,000 which showed a similar rate in comparison with IEI patients with non-hematologic cancers, but remarkably lower than the mortality in IEI patients with hematologic malignancies (800 per 100,000, 10-fold higher mortality rate compared to cancers in immunocompetent population) [32,33]. Consistently our observation within the current cohort also showed a significantly better survival of cases with non-hematologic malignancy than hematologic cancers.

The promise of individualized management of cancer patients relies on the discovery of actionable genetic alterations and the evaluation of novel targeted modalities. Despite this important practical usage, identification of a pattern of gene expression and epigenetics may help for elucidation of underlying pathogenesis and in-directly suggest potential targeted therapies. The most prevalent genes with somatic mutations identified in population-based cancer databases (mainly with non-hematologic cancers detected in immunocompetent individuals) were *TP53, KRAS* and *APC*. Of note, almost all of these top hints are immune-related genes and their germline mutation can be associated with IEI [3,34]. The current cohort of IEI patients with non-hematologic cancers and solved molecular diagnosis showed defects in DNA repair pathway (ATM deficiency), T cell development (AIRE deficiencies and STAT1 gain-of-function), B cell development (IKZF1 and PI3KR1 deficiencies), and innate immune pathway (IL12RB1 deficiency). Especially our finding highlighted mutations in *ATM* genes were associated with a high level of mortality and presentation of both hematologic and non-hematologic cancers in children and young-adult IEI patients. Within the four main categories of IEI associated with non-hematologic cancer in our study, several lines of evidence have been highlighted for the underlying mechanisms of tumorigenesis. Combined immunodeficient and antibody-deficient patients can develop several different cancer hallmarks including avoiding immune destruction, resisting cell death, inducing angiogenesis, deregulating cellular energies, activating invasion and metastasis, tumor-promoting inflammation, and enabling replicative immortality. Moreover, cases with syndromic CID due to DNA repair can increase genome instability and mutation and evade growth suppression (Figure 3) [9].

A wide range of ages at the last visit was recorded in our study, however, the median age observed was consistent with the previous report of the Immunodeficiency Cancer Registry (ICR) [35]. The male predominance in both hematologic and non-hematologic patients was in line with other studies [36]. The result can be justified by the fact that many types in the hematologic group were due to X-linked transmission disease including BTK deficiency. However, we could not observe any gene with X-linked inheritance in patients with non-hematologic cancers and the role of hormonal and other gender-specific environmental factors may play a role in this group of IEI patients. Moreover, the population with available genetic sequencing in this study were limited and prevents final conclusion over the entire population, indicating the need for investigation of genetic defects of patients with solid tumors in other IEI international registries in future studies.

## 5. Conclusions

Our data indicated that different types of genetic defects are underlying non-hematologic malignancies of IEI patients with higher prevalence compared to the normal population. Therefore, increased awareness of immunologists and oncologists about the risk of these types of malignancies in IEI patients, required regular monitoring and the performance of genetic evaluation for prognostic estimation and probable personalized treatment are essential. Although very few studies address these types of malignancies in the IEI cohorts, future studies and a better understanding of all molecular mechanisms of this phenotype ultimately improve management and, eventually, the survival of patients can be achieved.

## Figures and Tables

**Figure 1 cancers-15-00764-f001:**
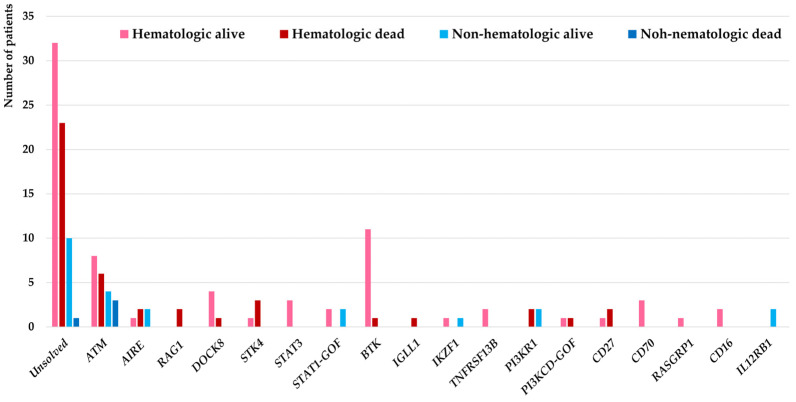
Molecular pathogenesis of 144 IEI patients associated with hematologic and non-hematologic malignancies and their impact on the mortality of patients.

**Figure 2 cancers-15-00764-f002:**
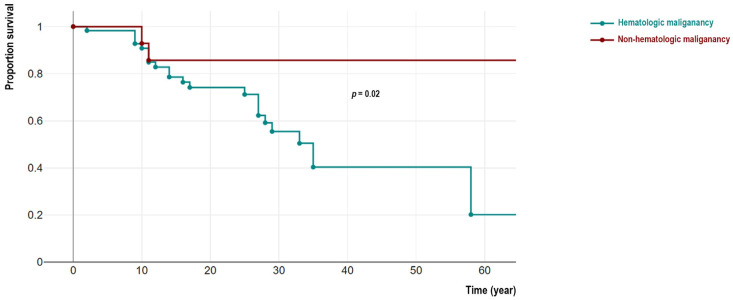
Comparison of survival analysis of cohort of IEI patients with hematologic and non-hematologic malignancies.

**Figure 3 cancers-15-00764-f003:**
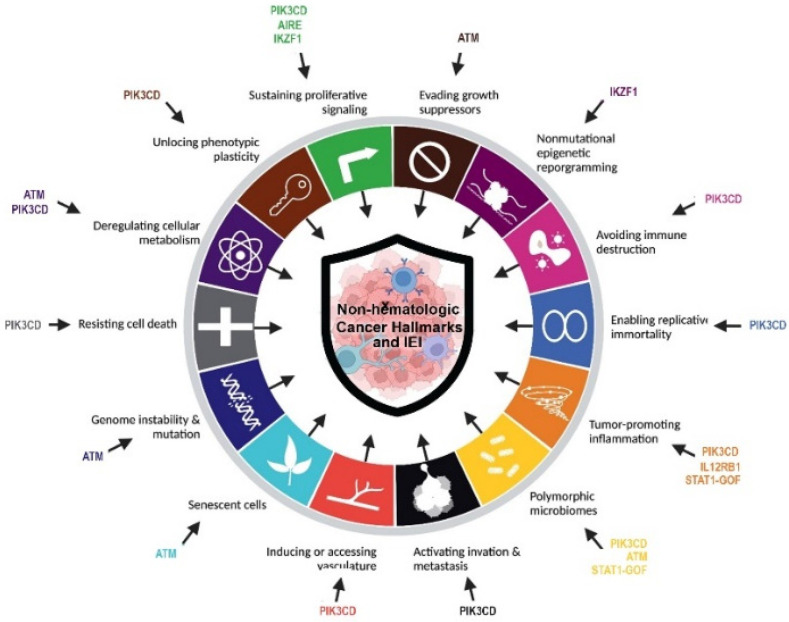
Cancer hallmarks in non-hematologic cancers of IEI patients according to the underlying discovered monogenic defects.

**Table 1 cancers-15-00764-t001:** Comparison of clinical and immunologic data of IEI patients with non-hematologic and hematologic tumors.

Parameters	Hematologic Cancers (*n* = 117)	Non-Hematologic Cancers (*n* = 27)	*p*-Value
Sex ratio (M/F)	66/51	18/9	0.32
Parental consanguinity (%)	77 (65.8)	20 (74.0)	0.40
Mortality (%)	43 (36.7)	5(18.5)	0.07
Median age at IEI onset, year (IQR)	3.5 (1.5–8.0)	5.0(2.5–12.2)	0.22
Median age at the diagnosis of malignancy, year (IQR)	13.0 (6.0–20.5)	19.2 (12.0–28.0)	0.08
Median age at the time of study *, year (IQR)	17.0 (10.9–24.2)	27.4 (22.7–36.4)	0.01 *
Otitis media (%)	39 (33.3)	7 (25.9)	0.27
Sinusitis (%)	43 (36.7)	12 (44.4)	0.28
Pneumonia (%)	60 (51.3)	13 (48.1)	0.76
Bronchiectasis (%)	23 (19.6)	11 (40.7)	0.02 *
Severe infections (%)	4 (3.4)	0	0.98
Chronic enteropathy (%)	14 (11.9)	11 (40.7)	<0.001 *
Failure to thrive (%)	27 (23.0)	7 (25.9)	0.75
Autoimmunity (%)	18 (15.3)	4 (14.8)	1.00
Allergy and atopic diseases (%)	26 (22.2)	4 (14.8)	0.39
Lymphoproliferation (%)	37(31.6)	6 (22.2)	0.33

* The age of last visit for deceased patients.

**Table 2 cancers-15-00764-t002:** Immunologic profiling of IEI patients with non-hematologic and hematologic tumors at the time of IEI diagnosis.

Parameters	Hematologic Cancers (*n* = 117)	Non-Hematologic Cancers (*n* = 27)	*p*-Value
Leukocyte/uL, Median (IQR)	6725 (4185–9074)	8237 (4421–11097)	0.51
Neutrophils/uL, Median (IQR)	3572 (2844–6820)	3586 (2804–4502)	0.86
Lymphocyte/uL, Median (IQR)	2039 (1517–3506)	1548 (609–2467)	0.07
Absolute CD3+ T cells/uL, Median (IQR)	1504 (757–2011)	472 (349–674)	0.11
Absolute CD3+CD4+ T cells/uL, Median (IQR)	417 (201–1154)	335 (197–726)	0.19
Absolute CD3+CD8+ T cells/uL, Median (IQR)	1533 (779–1845)	512 (379–670)	0.06
Absolute CD16/CD64+ NK cells/uL, Median (IQR)	249 (87–688)	165 (67–228)	0.35
Absolute CD19+ B cells/uL, Median (IQR)	192 (69–364)	39 (25–117)	0.09
Serum IgG mg/dL, Median (IQR) *	317 (84–603)	448 (89–821)	0.72
Serum IgA mg/dL, Median (IQR)	26 (13–47)	21 (9–32)	0.88
Serum IgM mg/dL, Median (IQR)	44 (22–115)	65 (32–106)	0.42
Serum IgE IU/mL, Median (IQR)	23 (5–80)	34 (7–96)	0.26

* The IgG levels were evaluated before initiation of immunoglobulin replacement therapy in IEI patients.

**Table 3 cancers-15-00764-t003:** Details of genetic studies of patients with 16 IEI patients with non-hematologic malignancies and defined molecular defects.

ID	Age (Year)	Sex	Clinical Diagnosis	IUIS Classification	Type of Cancer/s	Molecular Diagnosis	Mutation	Type of Mutation	Outcome
P1	10	M	Ataxia telangectasia	syndromic CID	Brain tumor + leukemia	*ATM*	Hom p.E1622X	Stop-gain	Death
P2	11	M	Ataxia telangectasia	syndromic CID	Mandibular squamous cell carcinoma + NHL	*ATM*	Hom p.L1851fsX1856	Frameshift	Death
P3	10	F	Ataxia telangectasia	syndromic CID	Squamous cell carcinoma of tongue + NHL	*ATM*	Hom p.Q1862RfsX25	Frameshift	Death
P4	10.5	F	Ataxia telangectasia	syndromic CID	Ovarian cystadenoma	*ATM*	Hom p.Tyr2371X	Stop-gain	Death
P5	13	M	Ataxia telangectasia	syndromic CID	Brain tumor	*ATM*	Hom c.2921+1G>T	Splicing	Alive
P6	9	M	Ataxia telangectasia	syndromic CID	Gastric adenocarcinomas	*ATM*	Hom del EX61-EX62	Large deletion	Alive
P7	22	F	Ataxia telangectasia	syndromic CID	Invasive ductal carcinomas	*ATM*	Hom. p.D2016G	Missense	Alive
P8	28	M	Common variable immunodeficiency	PAD	Gastric adenocarcinomas	*AIRE*	Het p.A500PfsX21	Frameshift	Alive
P9	64	M	Common variable immunodeficiency	PAD	Gastric adenocarcinomas	*AIRE*	Het p.R139X	Stop-gain	Alive
P10	20	M	Common variable immunodeficiency	PAD	Gastric adenocarcinomas	*IKZF1*	Het p.R143W	Missense	Alive
P11	25	M	hyper IgM syndrome	PAD	Colorectal cancer	*PI3KR1*	Het c.1425+1G>A	Splicing	Alive
P12	12	F	Common variable immunodeficiency	PAD	Gastric adenocarcinomas	*PI3KR1*	Het c.1425+1G>A	Splicing	Alive
P13	35	M	Mendelian susceptibility to mycobacterial diseases	innate immunity	Thyroid cancer	*IL12RB1*	Hom c.783+1G>A	Splicing	Alive
P14	37	F	Mendelian susceptibility to mycobacterial diseases	innate immunity	In situ ductal carcinoma	*IL12RB1*	Hom p.R212Q	Missense	Alive
P15	67	M	Chronic mucocutaneous candidiasis	innate immunity	Squamous cell carcinoma	*STAT1*	Het p.R274Q	Missense	Alive
P16	41	F	Chronic mucocutaneous candidiasis	innate immunity	Squamous cell carcinoma	*STAT1*	Het p.R274Q	Missense	Alive

## Data Availability

Data are contained within the article.

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
