# Peer review of "Clinical and Immunologic Characteristics of Non-Hematologic Cancers in Patients with Inborn Errors of Immunity"

_cancers, 2023, doi:10.3390/cancers15030764_

Round 1
Reviewer 1 Report
This paper describes interesting data but in some points it needs a more clear description and definition.
In general English writing is not always fluent. The final population analyzed is not numerous, so I think that for some point of the study is better a description of the results than a statistical analysis between two groups not equivalents.
Introduction line 56: immune destruction is not a sufficient explication of actions of immune system against cancer.
Materials and methods:
The population with available genetic sequencing is based on patient consensus to analysis, but this inclusion criteria prevents statistical considerations over the entire population and the final consideration must explicit this problem.
The description of the populations lacks a correlation between every molecular profile and the phenotype and/or the diagnosis of immunodeficiency. Due to the limited number of patients with solid tumors, I think the authors can explicit this with a table.
The 3 patients with both hematological and non-hematological malignancies deserve a separate analysis and for me is a bias to include them in the second group.
Results:
In table 1 the age at the last visit is not a clear data, may be better age at the time of study and may be that the major age is correlated with the major possibility of cancer onset.
How many patients with ATM deficiency are ataxia-telangiectasia patients?
Chronic enteropathy is correctly highlighted and must be correlated with frequency of gastro-intestinal cancers.
Discussion:
What means prescription at line 237?
The phrase about Pan-Cancer patients at line 252 in not clear
Author Response
This paper describes interesting data but in some points it needs a more clear description and definition.
First of all, we are grateful for your thoughtful and valuable comments. It is our pleasure that you have considered the provided points of the manuscript important. We rephrased and modified the new version based on all of the comments you mentioned. All of the changes in the revised manuscript have been highlighted.
- In general English writing is not always fluent.
Our response: We have edited the manuscript in this manner and the modified sections have been highlighted.
- Introduction line 56: immune destruction is not a sufficient explication of actions of immune system against cancer.
Our response: Indeed, we agree with this comment as this terminology is commonly used by cancer hallmarks papers, we have used it, but now we have added explanations regarding this terminology (Page2, line 57). We have also indicated other associated mechanisms and hallmarks that failure of immune response can present leading to cancer growth (Page 2, lines 58-63).
- The final population analyzed is not numerous, so I think that for some point of the study is better a description of the results than a statistical analysis between two groups not equivalents. The population with available genetic sequencing is based on patient consensus to analysis, but this inclusion criteria prevents statistical considerations over the entire population and the final consideration must explicit this problem.
Our response: Although the number of patients is limited appropriate statistical tests have been implemented by testing the normal distribution of gathered parameters and if required non-parametric evaluation have been performed accordingly (Page 4, lines 129-131), we have clarified this point mentioned by the reviewer as the limitation of the study in the discussion (Page 10, lines 300-302).
- The description of the populations lacks a correlation between every molecular profile and the phenotype and/or the diagnosis of immunodeficiency. Due to the limited number of patients with solid tumors, I think the authors can explicit this with a table.
Our response: A new table 3 has been added to compile the molecular diagnosis of patients with non-hematologic cancers and their associated tumor presentations based on this comment (Page 7).
- The 3 patients with both hematological and non-hematological malignancies deserve a separate analysis and for me is a bias to include them in the second group.
Our response: Based on this comment a separate clinical description of these three patients (P1-P3) has been presented to clarify this point in the newly generated Table 3 in the revised version of the manuscript.
- In table 1 the age at the last visit is not a clear data, may be better age at the time of study and may be that the major age is correlated with the major possibility of cancer onset.
Our response: We have modified this phrase in table 1 according to this suggestion and the age of cancer onset correlation with age at the time of study has been evaluated (Page 4 lines 157-159 and Table 1).
- How many patients with ATM deficiency are ataxia-telangiectasia patients?
Our response: All cases with ATM deficiency in our cohort presented with clinical presentation of ataxia-telangiectasia. This point has been clarified in the text now (Page 6, lines 189-190).
- Chronic enteropathy is correctly highlighted and must be correlated with frequency of gastro-intestinal cancers.
Our response: The correlation of chronic enteropathy with GI cancer has been presented in the revised version of the manuscript based on this comment (Page 4, lines 153-155).
- What means prescription at line 237?
Our response: We have modified this miss spelling phrase to “predisposition” (Page 8, line 255).
- The phrase about Pan-Cancer patients at line 252 in not clear
Our response: Thanks for these great comments, we have clarified now that we want to indicate the most frequent somatic mutations observed in the population-based databases of cancers are indeed immune-related genes (Page 9, line 270-273).
Reviewer 2 Report
Dear Editor,
Thanks for giving me the opportunity to review this interesting manuscript.
In their work, Delavari and Colleagues explore the prevalance of non-hematological cancers among patients with inborn errors of immunity and compared with hematological cancer in IEIs. The authors analyzed 5384 IEI patients registered in the Iranian national registry, evaluating clinical and immunologic phenotypes of patients with non-hematologic cancers, which accounted for roughly 20% of total cancer + IEI patients. Genetic analysis yelded heterogeneuos results, suggesting a wide spectra of mechanism beyond cancer development in IEI patients.
Please, find attached some comments that could improve the overall quality of the manuscript
- lines 43, 84, 105. the most recent IUIS classification should be used, older versions should be avoided (please correct reference 3 and 25 using only reference 38: https://doi.org/10.1007/s10875-022-01289-3 )
- line 105. the listed references look redundant. please avoid unnecessary self-citations
- line 115. the IUIS classification does include ten categories, not nine; bone marrow failure is missing. please correct
- line 166. the patient with HIGM what genetic defect does harbor? I don't see nor CD40-L or CD40 as well as UNG or AICDA in the list of genes.
- line 170-173. Table 2: is this immunological evaluation at IEI diagnosis? or at cancer diagnosis? or in another timepoint? how many pts were on IgRT (please specify as you present data on IgG)?
- line 179. did you performed genetic analysis in all the IEI patients with cancer diagnosis? in M&M you stated that target seq + WES were conducted on selected patients. it would be better for the readers to clarify how many pts underwent genetic analysis
- figure 1: is the Y axis percentage? please specify
- lines 180-198. did all the pts present pathogenic variants in the described genes? or some of them dysplayed VUS/SNP or novel variants? what about hz carriers of IEI-related genes? i think that genetic data should be presented in a more precise way, and that genetic diagnosis should be linked to the type of cancer that the patients presented (for example, the authors should make a table listing all the mutations & type of tumor for each patient)
- line 205. is it possible to know the causes of death of these patients? at least it would be very interesting to know if death was caused by cancer, cancer-related treatment's complications, other causes
Author Response
Thanks for giving me the opportunity to review this interesting manuscript. In their work, Delavari and Colleagues explore the prevalance of non-hematological cancers among patients with inborn errors of immunity and compared with hematological cancer in IEIs. The authors analyzed 5384 IEI patients registered in the Iranian national registry, evaluating clinical and immunologic phenotypes of patients with non-hematologic cancers, which accounted for roughly 20% of total cancer + IEI patients. Genetic analysis yelded heterogeneuos results, suggesting a wide spectra of mechanism beyond cancer development in IEI patients.Please, find attached some comments that could improve the overall quality of the manuscript
We appreciate your valuable comments. We are delighted to see your positive impression of this manuscript. We have edited all of the comments you mentioned and highlighted them in the revised text.
1- lines 43, 84, 105. the most recent IUIS classification should be used, older versions should be avoided (please correct reference 3 and 25 using only reference 38: https://doi.org/10.1007/s10875-022-01289-3)
Our response: We have updated our reference list based on this comment regarding the IUIS classification paper.
2- line 105. the listed references look redundant. please avoid unnecessary self-citations.
Our response: The reference list has been revised to concise the appropriate citations required for the provided text and methods and materials based on this comment (Pages 10-12).
3- line 115. the IUIS classification does include ten categories, not nine; bone marrow failure is missing. please correct.
Our response: We have revised the IUIS classification entities according to the latest update (Pages 4, line 124).
4- line 166. the patient with HIGM what genetic defect does harbor? I don't see nor CD40-L or CD40 as well as UNG or AICDA in the list of genes.
Our response: Based on this important note, we have generated a new table 3 to clarify this point as this patient with clinical phenotype of HIgM had genetic diagnosis of PI3KR1 deficiency (Page 7).
5- line 170-173. Table 2: is this immunological evaluation at IEI diagnosis? or at cancer diagnosis? or in another timepoint? how many pts were on IgRT (please specify as you present data on IgG)?
Our response: Based on this comment we have now clarified that the immunologic profiling is at the time of IEI diagnosis and also we clarified that all IgG levels were before Ig replacement therapy (Page 5).
6- line 179. did you performed genetic analysis in all the IEI patients with cancer diagnosis? in M&M you stated that target seq + WES were conducted on selected patients. it would be better for the readers to clarify how many pts underwent genetic analysis.
Our response: We have specified the number of patients evaluated and the diagnostic yield accordingly in the result section (Page 6, lines 181-183).
7- figure 1: is the Y axis percentage? please specify
Our response: We have clarified the axis unit in figure 1 (Page 6).
8- lines 180-198. did all the pts present pathogenic variants in the described genes? or some of them dysplayed VUS/SNP or novel variants? what about hz carriers of IEI-related genes? i think that genetic data should be presented in a more precise way, and that genetic diagnosis should be linked to the type of cancer that the patients presented (for example, the authors should make a table listing all the mutations & type of tumor for each patient).
Our response: As indicated in the method we only considered mutations with complete ACMG criteria with complete IEI inheritance for this study as the VUS and variants in incomplete Mendelian inheritance need further functional assays to be considered pathologic (Page 3, lines 110-114). The list of mutations associated is listed based on the comment in the new additional Table 3 (Page 7).
9- line 205. is it possible to know the causes of death of these patients? at least it would be very interesting to know if death was caused by cancer, cancer-related treatment's complications, other causes.
Our response: Thanks for these valuable comments, the total causes of death as well as cancer-caused mortality have been reported based on this comment (Page 8, lines 224-228).